# Formula Feeding Is Associated with Rapid Weight Gain between 6 and 12 Months of Age: Highlighting the Importance of Developing Specific Recommendations to Prevent Overfeeding

**DOI:** 10.3390/nu15184004

**Published:** 2023-09-15

**Authors:** Jigna M. Dharod, Kristen S. McElhenny, Jasmine M. DeJesus

**Affiliations:** 1Department of Nutrition, School of Health and Human Sciences, University of North Carolina at Greensboro, Greensboro, NC 27412, USA; k_stark@uncg.edu; 2Department of Psychology, College of Arts and Sciences, University of North Carolina at Greensboro, Greensboro, NC 27412, USA; jmdejes2@uncg.edu

**Keywords:** infancy, milk feeding, rapid weight gain

## Abstract

We examined differences in mean daily calorie intake and rapid weight gain risk among 6- to 12-month-old infants by milk feeding status: breastmilk only, breastmilk and formula (combined), or formula only. Another objective was to determine what frequency and amount of formula fed were associated with overfeeding among infants. Mother–infant dyads (*n* = 240) were recruited from a pediatric clinic mainly serving Medicaid recipients. At 6, 9, and 12 months of infants’ age, 24 h feeding recalls were conducted using the multiple-pass method. Infant weight measurements were accessed from clinic records to estimate rapid weight gain between 6 and 12 months. Among the participants, 82% received WIC. More than half of the participants were either African American or Latino by race/ethnicity. Calorie intake among formula-only fed infants was higher than in the other two milk feeding groups. One-fourth of the infants were experiencing rapid weight gain, and the risk was 3-fold higher among formula-only fed infants. Exceeding daily calorie requirements or overfeeding was associated with both formula amount and the frequency of feeding (*p* < 0.01). Specific guidelines and education on formula feeding practices are critical to prevent accelerated growth among infants. Gaining further understanding on parenting style and formula feeding practices is also warranted.

## 1. Introduction

Rapid weight gain during infancy is associated with overweight/obesity in childhood and later in life. Because the risk for overweight/obesity increases signficantly among children when they gain weight rapidly during infancy, preventing accelerated growth during infancy is recognized as an effective strategy against obesity [1,2]. Weight gain during infancy is more closely related to energy intake than at any other life stages; hence, feeding practices during infancy are key to predicting growth trajectories [3]. 

During the first 6 months of infancy, all calories are expected to come from milk feeding, either from breastmilk and/or formula. Even in late infancy (6 to 12 months), up to 50% of daily calories are expected to be met by milk feeding [4]. Therefore, breastmilk and formula feeding practices play a significant role in predicting weight gain during infancy. 

While the benefits of breastfeeding are recognized, in the U.S., more than half of infants receive formula by 3 months of age, with higher rates among low-income, minority groups [5]. Formula-fed infants gain more weight than breastfed infants, mainly attributed to the higher protein content in formula than in breastmilk. For instance, protein content in formula ranges from 1.3 to 1.9 g/100 mL (versus approximately 1.03 g/100 mL in breastmilk) [6]. In several clinical trials, infants who were fed low-protein formula grew more slowly and demonstrated a growth pattern such as that of breastfed infants [7]. However, there is a dearth of information about differences in calorie intake among breastmilk-fed and formula-fed infants, and to what extent overfeeding is occurring among the latter. More information on calorie intake from formula and how it varies by frequency and amount can help determine the overfeeding risk and set up an upper intake level for daily formula intake. 

Current formula-feeding guidelines are focused on procedural and safety recommendations, such as sterilization, preparation, and storage of the bottle (i.e., to prevent bacterial growth). Although these guidelines are important, parents lack specific guidelines on feeding itself, such as the upper limit to prevent overfeeding. A recent review indicated that formula packaging providing information on formula preparation steps remain the main information. Specifically, information on how much to feed in each episode, the general frequency of feeding and guidance on how to recognize infant satiety cues are lacking [8].

The Special Supplemental Nutrition Program for Women, Infants and Children (WIC) serves approximately 1.5 million infants living below 185% of the federal poverty level in the U.S. Approximately 75% of infants enrolled in WIC are either partially or fully formula-fed at any given time [9]. Low-income families were the most affected by the recent formula shortage, as almost half of all formula in the U.S. is purchased by families enrolled in WIC [10]. However, how formula feeding practices affect daily calorie intake among infants is not fully examined. This is critical since WIC households are at a higher risk of experiencing food and nutrition insecurity. In food insecurity, caregivers often experience food shortage and high sensitivity to food. It is also associated with an indulgent parenting style and viewing fussiness as a sign of hunger [11]. WIC’s fully formula-feeding package is designed to provide enough formula to meet the energy needs for the infants. However, it was found that WIC households often run out of formula at month’s end, indicating potential overfeeding [12]. Hence, in alignment with the call for action to promote nutrition security and address health disparities [13], the objectives of this study were to (1) examine differences in mean daily calorie intake and rapid weight gain risk among 6- to 12-month-old infants by milk feeding status: breastmilk only, breastmilk and formula (combined), or formula only; and (2) determine what frequency and amount of formula fed were associated with overfeeding among infants living in low-income households. 

## 2. Materials and Methods

This study involving 240 mother–infant dyads was approved by the Institutional Review Board at the University of North Carolina at Greensboro. We recruited mother–infant dyads from a local pediatric clinic primarily serving families of low socioeconomic status. Selection criteria for participation included (1) mother at least 18 years of age; (2) carried a singleton pregnancy; and (3) delivered at least at 37 weeks of gestational age or longer. Infants with any health issue likely affecting growth, such as endocrine issues, were not included in this study. 

To estimate the necessary sample size, Monte Carlo simulations were conducted using potential effect size estimates of 0.2 to 0.5. Standards for minimum coverage were set at 95% with estimated bias at less than 5%, and power ≥ 0.8. Accounting for attrition rate of 20%, a minimum sample estimated was 200 mother–infant dyads for a power of 0.99 to 0.88. However, with the clinic’s support, we continued recruitment beyond the minimum sample target during the study period. 

The research assistants approached mother–infant dyads in the waiting room at their 2 month or younger wellness visits at a local pediatric clinic to explain this study and assess eligibility. Once interest was indicated and the dyads passed eligibility requirements, mothers provided written consent for study participation involving interviews at regular intervals. Mothers were also asked to provide HIPPA permission to access infants’ medical records to access their weight measurements taken at regular intervals. Upon recruitment, when infants were 6, 9, and 12 months old, mothers were contacted via phone to collect sociodemographic information and carry out 24 h feeding recalls. Mothers received a grocery store gift card incentive upon completion of each recall. 

The 24 h feeding recalls were conducted using Nutrition Data System for Research (NDSR), a computer-based software application developed at the University of Minnesota Nutrition Coordinating Center. The NDSR facilitates the collection of recalls in a standardized fashion using the multiple-pass method. The NDSR database includes calorie and nutrient values of approximately 8000 brand products, including different infant formula brands, fruit and vegetable pouches, and jarred foods. 

To facilitate accurate portion size estimation, one day prior to each recall, pictures of standard sippy cups, formula bottles, and spoons were texted to participants. Based on participant preference, the recall was conducted in either English or Spanish. Prior to conducting the recalls for this study, the research team of graduate and undergraduate research assistants participated in a 2-day NDSR training session and completed 10 practice 24 h recalls.

Infant weights, measured at well-child visits by trained clinic staff, were retrieved from clinic records to control for baseline weight and calculate weight gain over time. For this study, rapid weight gain was defined as a change of >0.67 standard deviations in the weight-for-age z-score between 6 and 12 months of infant’s age [14]. 

Of the 240 participants, 30 were lost to follow up at 12 months due to moving out of the study area and discontinuing with the clinic. Additionally, 22 had already switched to regular milk before the 12-month interview and were not included in the analyses. Hence, the sample size for formula feeding practices at 12 months and rapid weight gain analyses was 188, 80% of the original sample. 

### 2.1. Interviews and Measures

Sociodemographic variables: Information about maternal education, marital status, household size, and income was collected. Questions were also asked to determine participation in food assistance programs, such as WIC. Mothers were asked to report their height and current body weight to estimate their BMIs. Self-reported height and weight have been used and shown to be accurate to group people correctly into BMI categories of normal vs. overweight/obesity [15,16]. Information on race/ethnicity was also collected. 

The 24 h feeding recalls: At 6, 9, and 12 months, feeding recalls were carried out using the multiple-pass method [17]. This method involves first noting times of feeding and reviewing with the participant, then collecting detailed descriptions of each feed including brand or type of food with the amounts eaten, and finally reading out loud the whole recall to the participant to ensure no feeding was missed. For each feeding, detailed information on food, amount, and preparation steps were collected. Questions were also asked to note what amount of food prepared was consumed by the infant. For instance, in the case of formula feeding, first information on the brands and types of formula was collected (powdered, ready to use, concentrate, etc.). For powdered formula (the most common type), we asked about the amount of powdered formula (in scoops), amount of water added (in ounces), and how much of the prepared amount the infant consumed to adjust and accurately measure total calorie intake.

When recording direct breastfeeding, we estimated milk intake using data from the literature [18,19,20]. For the 6-month recalls, the total breastmilk and formula volumes were adjusted to 675 mL/day; for the 9 and 12 month recalls, the total breastmilk and formula volumes were adjusted to 600 mL/day to account for more calories expected from complementary foods. No adjustments were made for pumped breast milk feeding, since participants could report the specific amount. Information on other foods and solids were collected using specific brand and portion size. 

Prior to analysis, quality checks were conducted on the 24 h recalls identifying any errors in reporting or data entry specifically affecting portion size and thereby total calorie intake. Recalls that involved feeding ‘adult’ foods (such as pizza) sometimes involved adjustments. For instance, if the mother reported feeding some portion of a pizza slice, then the portion was estimated based on approximate bites eaten. Based on what was fed with complementary foods, infants were grouped into three categories of milk-feeding type: breastmilk only, breastmilk and formula (combined), or formula only. The NDSR nutrient and food group output files were used to estimate total calorie intake and what number of total calories came from: milk feeding (breastmilk/formula) or other solid and liquid foods (referred to as complementary foods).

### 2.2. Statistical Analyses

Analyses were conducted using the SPSS software (IBM SPSS Statistics for Windows, Version 21.0. Armonk, NY, USA: IBM Corp.). The statistical significance threshold was set at *p* < 0.05. Descriptive statistics were computed to examine sociodemographic characteristics, rate of rapid weight gain, and feeding practices at 6 and 9 months of age. To examine differences in daily calorie intake by milk feeding type, an analysis of variance (ANOVA) was carried out with post hoc Tukey tests to examine pairwise differences between groups. 

Binary logistic regression was conducted to estimate the risk for rapid weight gain by milk feeding type after controlling for key covariates associated with weight gain trajectory among infants [1]. The covariates included in the model were: parity, type of delivery, maternal age, education, marital status, BMI, household income, infant weight at 2 months, and whether infant was fed any solids prior to 4 months of age. Except for infant 2-month weight, covariates were entered as categorical variables with a specific reference value. Multicollinearity was tested to ensure it did not affect odds estimation significantly. Lastly, ANOVA was carried out to examine differences in frequency and amount of formula fed between those who were within the daily energy requirement versus those who were not. The daily calorie requirements for 6-, 9-, and 12-month-old infants were based on calculations estimated specifically for formula-fed infants [21]. 

## 3. Results

Of the 240 mother–infant dyads, approximately two-thirds of the participants were either African American or Latino (Table 1). Over half of the mothers had graduated from high school or earned a GED. Thirty-eight percent of the mothers were employed full-time or part-time. The majority received WIC assistance and 45% participated in the Supplemental Nutrition Assistance Program (SNAP). 

At 6 months, over half of the infants were fed formula; the remaining infants were fed breastmilk or a combination of breastmilk and formula (Table 2). At 9 months, nearly two-thirds of infants were given formula, which persisted at the 12-month follow up. At 6 months, infants in the formula only group were consuming significantly more calories than the other two groups (i.e., breastmilk only or breastmilk and formula (combined) milkfeeding groups). At 9 months, a significant difference in total calorie intake was seen between the formula only and breastmilk only feeding groups. No statistically significant differences in total calorie intake were seen at 12 months. As shown in Table 2, calories from complementary food did not differ by milk feeding group at any of the three time points. Differences in total calories from milk feeding were statistically different at each time point (Table 2).

Rapid weight gain between 6 and 12 months was seen among 24% of the infants. In bivariate association, as shown in Table 3, rapid weight gain was more common among primiparous versus multiparous infants. No significant difference in rapid weight gain was seen by other demographic characteristics, such as maternal age, education, type of delivery, and race/ethnicity.

In multivariate analyses estimating rapid weight gain risk, it was found that milk-feeding status at 6 months was significant predictor after controlling for key sociodemographic variables, including early introduction to solids. Milk feeding status of formula only at 6 months was associated with a 3-fold increased risk for rapid weight gain between 6 and 12 months (Table 4).

Lastly, the frequency of feeding and amount of formula consumed were compared between those who were within versus above the recommended daily calorie requirement at 6, 9, and 12 months. (Table 5). The recommended intake at 6 months is 700 kcal total which increases to 750 kcal at 9 months and 850 kcal at 12 months. Overall, the frequency of formula consumption decreased with an increase in age (i.e., between 6, 9 and 12 months). In comparison, the frequency of formula feeding was significantly higher among infants crossing the daily calorie requirements than their counterparts at 6 and 9 months. No significant difference by frequency was seen at 12 months. By amount, infants exceeding the daily recommended calorie intake were consuming approximately 50% more formula than their counterparts at all the three time points. Specifically at 6 and 9 months, exceeding daily calorie intake was associated with 30 ounces or more of formula feeding (Table 5). 

## 4. Discussion

Infant formula is designed to mimic the nutritional composition of breastmilk and it is an effective substitute when breastfeeding is not feasible due to low milk production of the mother, infant sucking or latching issues, medical conditions of infant or the mother, unwillingness to breastfed, or other barriers to breastfeeding. The growth patterns differ between breastmilk and formula-fed infants, with formula-fed infants expected to weigh on average 400–600 g more than breastfed infants at 12 months of age [22]. Our results demonstrate that formula overfeeding is common and infants who are receiving milk calories from formula only are at a higher risk of having accelerated growth versus those receiving breastmilk during complementary feeding phase.

A national study indicated that, on average, infants consume approximately 836 calories daily during the complementary phase of 6 to 12 months [23]. In our study, average daily calorie intake was in the same range, with an expected gradual increases in intake between 6, 9, and 12 months of age. By milk-feeding status, formula-fed infants were consuming significantly more calories than the breastmilkonly group (approximately 10% and 20% higher calories at 6 and 9 months, respectively). Specifically, these differences in calorie intake were not driven by complementary foods, since in comparison, calories from complementary foods were not significantly different between the breastmilk only vs. combined vs. formula only groups. The calorie differences were solely the result of calorie intake from milk feeding type. Based on Federal Nutrition Monitoring Data, the 2020 Dietary Guidelines Advisory Committees also examined calorie intake between infants fed breastmilk only vs. breastmilk and formula (combined) vs. formula only from 6 to 12 months. Aligning with our study, formula-fed infants in that study were consuming 33% more calories than breastmilk-fed infants [24].

In examining formula or bottle-feeding practices, the evidence suggests that infant control over intake is reduced, and parent feeding style becomes prominent. Variables such as infant positioning and attachment on the bottle, size of the bottle, using an opaque vs. clear bottle, and the nipple flow rate all affect intake [25,26,27]. Breast- and bottle-fed infants have similar suck–swallow–breath patterns [28]; hence, intake amounts in formula-fed infants have been mainly attributed to parenting style and bottle-feeding equipment (bottle and nipple). Prior studies show that when a parent has a pressurized feeding style, the infant consumes more milk [29], meaning parents end feeding when the bottle is empty or close to empty regardless of cues from infant. Infant temperament is also shown to play a role in predicting total intake in bottle feeding. In an observational study, it was found that infants with low orienting/regulation capacity consumed more when mothers were distracted. Such an interaction was not seen for infants with high regulation capacity [30]. In our study, expressed breastmilk feeding was not very common, but differences in breast milk intake between direct and expressed mode can also potentially affect daily calorie intake.

Our study shows that overfeeding is common and that formula-fed infants were more likely to exceed daily calorie requirements because of a significantly higher frequency and amount of formula feeding. Most of our participants were participating in WIC and were receiving formula through the program. In WIC’s fully formula feeding infant package, the amount of formula provided is enough to meet the daily recommended calorie intake. However, in a national WIC sample, more than one-third of participants reported that they were getting formula from another source to supplement their WIC supply. Among this group, infants consumed significantly more calories and weighed more at any given time compared to infants who were fed formula within the WIC allocation [12]. Similarly, in another study, approximately 43% of WIC infants were overfed or were consuming more than WIC-allocated amount of formula [31]. These findings and our study highlight the importance of establishing formula feeding guidelines to help educate parents and prevent overfeeding. 

To promote responsive feeding overall, the World Health Organization’s Global Criteria for the Baby-Friendly Hospital Initiative has been revised to provide support not only those who initiate breastfeeding but also to mothers who opt to do formula feeding [32]. It is possible that marketing and claims made by formula manufacturers affect parents’ perception that overfeeding of formula is not an issue and limiting formula feeding is not required. Specific recommendations, such as the proportion of formula to other complementary food and how to recognize infants’ cues to prevent overfeeding, are warranted. In our study, combined feeding was also common. aApproximately one-third of our participants were of Latino origin. In particular, a practice of combined feeding, commonly referred as ‘las dos cosas’, is very common among Latino community. Our previous study has shown that when the ratio of formula versus breastmilk feeding is higher, the risk for overfeeding increases [33]. 

Results of this study specifically indicate that the formula-only group was at higher risk of overfeeding and rapid weight gain, highlighting the importance of continuation of even some level of breastfeeding post 6 months. Evidence suggests that formula feeding increases daily added sugar intake significantly. For instance, in a study by Kong et al., to thetotal added sugar intake through formula was calculated among 9- to 12-month-old infants. Based on the 24 h feeding recalls, it was found that formula contributed to 66% of daily added sugar intake. A significant positive association was seen between added sugars from milk-based sources and risk for rapid weight gain after controlling for sociodemographic variables such as gestational age and maternal pre-pregnancy BMI. In comparison to breastfed infants, the amount of added sugar intake was 2-fold higher among formula-fed infants [34]. The formula food label does not list the added sugar amount; however, maltodextrin and polydextrose in the infant formula can contribute significantly to added sugar intake. 

To the best of our knowledge, this is the first study to identify a calorie intake pattern by milk feeding status during the complementary feeding phase. This study also provides information on the amount of formula consumed on average among infants and the extent to which it differs between those who stay within vs. exceed daily calorie requirements. However, we recognize some limitations. First, due to dropouts, the sample size at 12 months was lower than the actual sample size, causing rapid weight gain analyses and formula intake analyses with 80% of the original sample size. However, the multivariate model was robust in explaining variances and predicting rapid weight gain. Second, relying on a single 24 h recall at each time point limits the reliability of our findings. In addition, among our participants, direct breastfeeding was more common than feeding pumped breastmilk, and hence intakes were based on the estimations from previous literature on average milk intake versus measured amount. Finally, we studied a convenience sample from a mid-size pediatric clinic mainly serving low-income women, limiting the generalizability of our findings. Nonetheless, this study highlights the need for formula feeding guidelines, specifically education that can help parents with responsive feeding. In the future, examination of formula feeding practices and appetite development among infants is warranted to fully understand the long-term impact and importance of education on optimal feeding practices. 

## Figures and Tables

**Table 1 nutrients-15-04004-t001:** Sociodemographic characteristics of study participants (*n* = 240).

Characteristics	Mean (SD)
Maternal age (years)	29 (6)
Household size	4 (2)
Household income (monthly)	$1990 ($1162)
	*n* (%)
Infant Sex	
Male	106 (44)
Female	134 (56)
Parity	
Primiparous	92 (39)
Multiparous	148 (61)
Type of delivery	
Vaginal	195 (79)
C-section	45 (21)
Maternal Race/Ethnicity	
African American	93 (38)
Latino	92 (38)
Non-Latino White	22 (10)
Other ^b^	33 (14)
Education	
Less than High School	49 (20)
High School/GED	135 (56)
Attended college	56 (24)
Marital Status	
Married or living with partner	104 (44)
Single/divorced/separated	136 (56)
Maternal BMI status	
Normal	41 (17)
Overweight/obese	199 (82)
Employment Status ^c^	
Employed (full/part time)	85 (38)
Unemployed	155 (64)
Participating in WIC	204 (82)
Participating in SNAP	110 (45)

Percentages and mean numbers are rounded. *n* = 180, others either reported no income, don’t know, or refused to answer the question. ^b^ “Other” group includes: American Indian/Alaska Native, Asian, and multiple race/ethnicity; ^c^ employment status at enrollment; WIC: Special Supplemental Nutrition Program for Women, Infants and Children; SNAP: Supplemental Nutrition Assistance Program.

**Table 2 nutrients-15-04004-t002:** Differences in total and complementary food calories by milk feeding type.

Infants’ Ages	6-Month (*n* = 240)				9-Month (*n* = 240)				12-Month (*n* = 188)			
	Formula Only	BM + Formula	BM Only	*p*	Formula Only	BM + Formula	BM Only	*p*	Formula Only	BM + Formula	BM Only	*p*
*n* (%)	140 (58)	49 (20)	52 (22)		154 (64)	36 (15)	50 (21)		120 (64)	21 (11)	47 (25)	
Total Calories (kcal)	775 ^ab^ ± 257	645 ^b^ ± 169	589 ^a^ ± 179	<0.001	916 ^a^ ± 261	818 ± 265	757 ^a^ ± 215	<0.001	967 ± 248	859 ± 283	887 ± 276	0.075
Calories from Milk feeding	700 ^ab^ ± 267	571 ^b^ ± 143	496 ^a^ ± 143	<0.001	646 ^ab^ ± 254	524 ^b^ ± 219	434 ^ab^ ± 59	<0.001	505 ^ab^ ± 243	468 ^b^ ± 185	367 ^a^ ± 133	<0.001
Calories from Complementary Food	75 ± 91	74 ± 113	93 ± 136	0.574	270 ± 202	294 ± 231	323 ± 206	0.298	462 ± 259	391 ± 293	520 ± 265	0.166

BM: Breastmilk. Same superscript for each time point denotes significant difference between the feeding groups based on the post-hoc Tukey HSD test, *p* < 0.005; in direct breastfeeding amount was approximated using the following average daily milk (formula and breastmilk) intake estimated for infants: 675 mL for 6-month, and 600 mL for 9- and 12-month. No adjustments were made for pumped breast milk feeding, since participants reported the specific amount.

**Table 3 nutrients-15-04004-t003:** Prevalence of rapid weight gain overall and by key sociodemographic characteristics. (*n* = 188).

Number of Infants Experienced Rapid Weight Gain from 6 to 12 Months	45 (24%)	
	*n* (%) ^a^	*p*
Maternal BMI		
Normal	7 (22)	0.748
Overweight/Obese	41(25)	
Maternal Age		
<30	24 (26)	0.863
≥30	25 (25)	
Education		
High School, GED or less	28 (27)	0.483
More than high school	22 (22)	
Employment status b		
Working (full/part-time)	28 (21)	0.328
Not working	22 (28)	
Race/Ethnicity		
African American	20 (28)	
Latino	17 (20)	0.620
Non-Latino White	4 (22)	
Other ^c^	8 (29)	
Parity		
Primiparous	26 (32)	0.032
Multiparous	23 (19)	
Type of delivery		
Vaginal	40 (25)	0.670
C-section	9 (22)	
Infant’s sex		
Male	27 (29)	
Female	22 (20)	0.120

^a^ Chi-square; Percentages rounded. ^b^ employment status during the time of recruitment ^c^ “Other”. group includes: American Indian/Alaska Native, Asian, and multiple race/ethnicity.

**Table 4 nutrients-15-04004-t004:** Risk factors for rapid weight gain in late infancy by 6- and 9-month feeding status (*n* = 188).

	Feeding Status at 6 Months			Feeding Status at 9 Months		
	OR	CI	*p* ^a^	OR	CI	*p* ^a^
Maternal BMI (Ref: Normal Weight)						
Overweight/obese	0.96	0.33, 2.76	0.944	1.09	0.36, 3.25	0.876
Maternal Age (Ref: <30)						
30 or more	1.07	0.50, 2.27	0.859	1.54	0.69, 3.44	0.289
Maternal Employment (Ref: unemployed)						
Employed part/full-time	1.08	0.47, 2.41	0.870	1.14	0.48, 2.67	0.756
Maternal Education (Ref: High School or less)						
More than high school	0.54	0.24, 1.20	0.133	0.52	0.24, 1.19	0.125
Ethnicity (Ref: non-Latino White)						
African American	0.98	0.26. 3.67	0.970	0.68	0.17, 2.70	0.590
Latino	0.74	0.17, 3.23	0.692	0.59	0.13, 2.63	0.492
Other	1.62	0.34, 7.60	0.539	1.87	0.39, 8.82	0.425
Parity (Ref: Multiparous)						
Primiparous	2.16	0.98, 4.63	0.056	2.16	0.98, 4.76	0.055
Type of delivery (Ref: Vaginally)						
C-section	0.81	0.32, 2.04	0.659	0.76	0.28, 2.01	0.585
Early introduction to solids * (Ref: No)						
Yes	1.59	0.68, 3.69	0.279	1.61	0.66, 3.93	0.288
Infant weight at 2 months (continuous)	0.99	0.98, 1.00	0.396	0.99	0.97, 1.00	0.267
Milk-Feeding Status ^‡^ (Ref: Breastmilk only)						
Combined	3.27	0.89, 12.01	0.074	1.33	0.31, 5.63	0.699
Formula only	3.42	1.03, 11.35	0.044	3.18	0.93, 10.90	0.065

^a^ Binary Logistics; OR: Odds Ratio; CI: Confidence Interval; Rapid weight gain was defined as a change of more than 0.67 standard deviations in weight-for-age z-score between 6 and 12 months. * Early introduction to solids refer to whether infant was fed any solids ≤4 months of age. ^‡^ along with complementary foods or other solids and liquids.

**Table 5 nutrients-15-04004-t005:** Differences in the frequency and amount by daily calorie requirements among formula-fed infants.

6-Month-Old Infants	Overall	≤700 kcal	>700 kcal	*p* ^b^
	*n* = 189 ^a^	*n* = 90	*n* = 99	
		mean (SD)		
Bottle feeding frequency	5.26	4.39 (1.89)	6.06 (1.94)	<0.001
Amount of formula fed (oz.)	27.79	19.61 (8.97)	35.44 (12.36)	<0.001
9-Month-Old Infants	Overall	≤750 kcal	>750 kcal	*p* ^b^
	*n* = 190 ^a^	*n* = 65	*n* = 125	
		mean (SD)		
Bottle feeding frequency	4.32	3.76 (1.34)	4.61 (1.68)	<0.001
Amount of formula fed (oz.)	26.13	18.60 (8.68)	30.05 (12.54)	<0.001
12-Month-Old Infants	Overall	≤850 kcal	>850 kcal	*p* ^b^
	*n* = 141 ^a^	*n* = 52	*n* = 89	
		mean (SD)		
Bottle feeding frequency	3.45	3.13 (1.37)	3.62 (1.54)	0.058
Amount of formula fed (oz.)	20.83	16.07 (7.57)	23.63 (11.46)	<0.001

For each age, daily calorie requirements are based on the following research paper: [21]. Bottle feeding represents formula feeding with/without addition of other foods. ^a^ Infants who were fed formula only or both formula and breastmilk combined along with complementary foods; ^b^ ANOVA.

## Data Availability

To prevent any breech in confidentiality, the dataset will not be shared.

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
