# Peer review of "Formula Feeding Is Associated with Rapid Weight Gain between 6 and 12 Months of Age: Highlighting the Importance of Developing Specific Recommendations to Prevent Overfeeding"

_nutrients, 2023, doi:10.3390/nu15184004_

Round 1
Reviewer 1 Report
I had the opportunity to review your paper titled " Formula feeding is associated with rapid weight gain between 6 and 12 months of age: Highlighting the importance of developing specific recommendations to prevent overfeeding." for the Nutrients journal.
Overall, I found your research to be well-conducted, and your results provide valuable insights that Infants on formula diets exhibit greater calorie intake, elevating their risk of rapid weight gain. Overfeeding, linked to 5+ daily formula feedings and 25+ oz and calls for guidelines and education on type of early life feeding practice to address infant obesity risk. Your conclusions are supported by sound data, and the language used throughout the paper is clear and engaging. However, some points of attention for your consideration:
1- It's essential to ensure that your references are cited correctly and consistently throughout the paper. For example, line 126 , the citations are wrong and reference 29 and 30 are not cited throughout your paper. In lines 36 and 44 , the references are also missing. Please review and edit your references to align them with the journal's citation guidelines.
2 - To estimate energy intake in breastfed and mixed fed infants, an estimated milk intake from literature is used. However, this is a fixed value for each infant which is applied without consideration of body size. It can be questioned whether this is a valid approach for concluding on statistically significant differences in caloric intake between formula fed and (any) breastfed infants, as presented in Table 2 (the variation in caloric intake in the BM and BM+formula groups seem to be fully reliant on supplementary food variation).
3- Please include the Calories from milk intake in Table 2 (and in the footnote the note on assumed intake in BM fed infants)
4- I believe that your study would benefit from including birth weight and the timing of the introduction of complementary feeding as covariates in your statistical analysis (line 151). These factors could potentially influence the relationship between infant feeding practices and rapid weight gain.
5- Which were your assumptions for required sample size, a power calculation is to be provided.
Reviewer 2 Report
Drs Dharod and colleagues present their study assessing breastfeeding and formula feeding and weight trajectories in infants,.
The study is important as it raises an important point with regards to longterm metabolic risks from early overfeeding.
The introduction provides sufficient information for non-US readers about the structure of the local programme and the baseline characteristics of the population.
The methods are described well and the statistical analysis is valid. The authors discuss the limitations and assumptions resulting from the limited time points of parent contact over time . The results presented support the discussion points.
style :
suggestion: L 264: The sample at 12 months reduced to 80 % of the size at enrollment Hence analysis of weight gain and formula intake were based on a smaller sample.
Reviewer 3 Report
This study thoroughly introduces the importance of comparing breastmilk and formula feeding. The author collected information using reasonable methods and selected appropriate inclusion criteria. The study results showed that infants fed with formula have a higher risk of rapid weight gain compared to those fed with breastmilk.
The author provides a comprehensive discussion of the study results, making them credible. Some optimization may be needed in presenting the results. For example, in table 3, only rates are shown, and presenting quantities would make it easier to understand. Additionally, some content in table 2 exceeds the page, resulting in some information loss.
Round 2
Reviewer 1 Report
Thank you for adequately addressing all concerns raised.